# Direct Synthesis of Gold Nanoparticles in Polymer Matrix

**DOI:** 10.3390/polym15010016

**Published:** 2022-12-20

**Authors:** Quang Truong Pham, Gia Long Ngo, Xuan An Nguyen, Chi Thanh Nguyen, Isabelle Ledoux-Rak, Ngoc Diep Lai

**Affiliations:** LuMIn, ENS Paris-Saclay, CentraleSupélec, CNRS, Université Paris-Saclay, 91190 Gif-sur-Yvette, France

**Keywords:** Au nanoparticles, plasmonics, polymer matrix, nanocomposite, thermal annealing

## Abstract

We report an original method for directly fabricating gold nanoparticles (Au NPs) in a polymer matrix using a thermal treatment technique and theoretically and experimentally investigate their plasmonic properties. The polymeric-metallic nanocomposite samples were first prepared by simply mixing SU-8 resist and Au salt with different concentrations. The Au NPs growth was triggered inside the polymer through a thermal process on a hot plate and in air environment. The Au NPs creation was confirmed by the color of the nanocomposite thin films and by absorption spectra measurements. The Au NPs sizes and distributions were confirmed by transmission electron microscope measurements. It was found that the concentrations of Au salt and the annealing temperatures and durations are all crucial for tuning the Au NPs sizes and distributions, and, thus, their optical properties. We also propose a simulation model for calculations of Au NPs plasmonic properties inside a polymer medium. We realized that Au NPs having large sizes (50 to 100 nm) play an important role in absorption spectra measurements, as compared to the contribution of small NPs (<20 nm), even if the relative amount of big Au NPs is small. This simple, low-cost, and highly reproducible technique allows us to obtain plasmonic NPs within polymer thin films on a large scale, which can be potentially applied to many fields.

## 1. Introduction

Metallic nanoparticles (NPs) are widely used in diverse applications, thanks to their surface plasmon resonance (SPR), i.e., the collective oscillation of free electrons in resonance with the incident light, for example, in photocatalysis [1,2], sensing [3,4,5], optical data storage [6,7,8], or medicine [9,10]. Many applications require the incorporation of these plasmonic NPs into different materials, especially polymers, to form the so-called plasmonic nanocomposites [11,12,13,14,15].

To fabricate the metal NPs inside a polymer medium, the common synthesis approach is to insert the presynthesized NPs, by using different techniques [16,17], into polymer aqueous solution. It was shown that, in this case, NPs usually aggregate and the metallic NPs lose their SPR properties. For this reason, stabilizing agents are required to obtain homogeneously dispersed NPs [18]. However, achieving a homogeneous spatial distribution at large scales inside the polymer matrix is still challenging [19]. Such disadvantages may directly affect the optical properties of the system [20], notably by inducing a large broadening of the plasmon resonance peak. To solve this problem, direct synthesis methods of NPs inside polymer matrix have been proposed. Different methods are investigated, such as chemical methods [21,22], and physical methods such as thermal treatment [13,14,23] and ultraviolet (UV) illumination [12,24]. The direct synthesis method thus allows to separate the polymer-shaping process from the control of NPs dispersion inside the medium [19].

Among different polymers, SU-8 resist is a favourable polymer because it is a low-cost material and is commonly applied in various fields, such as photonic devices [25,26] and microfluidics [27,28], thanks to its excellent mechanical properties [29], chemical stability [30], good biocompatibility [31], and functionalizability [32]. In addition according to previous work, SU-8 has great potential in photoreduction due to the generation of high chemical functionality and free radicals during the photochemical process [12]. There are several works regarding the fabrication of gold (Au) NPs inside SU-8 resist, mainly based on the photoreduction effect. For example, Shukla et al. presented a method for fabricating Au nanostructures within a polymeric matrix using the two-photon lithography technique [33]. In such a system, there is a simultaneous reduction of Au precursor and a polymerization of SU-8 resist, resulting in Au NP-doped polymeric lines. Chen et al. reported a method for directly synthesizing an Au NPs monolayer on the surface of SU-8 under a UV exposure [12]. Despiteall these publications, the formation of Au NPs inside SU-8 resist by light or by thermal effect has not been fully investigated yet.

In the present work, we report a simple, low-cost, fast, reproducible, and efficient method to directly fabricate Au NPs inside SU-8 based on thermal effects without any intervention of light. In contrast to other chemical and photoreduction methods, only SU-8 and Au salt (HAuCl${}_{4}$·3H${}_{2}$O) are needed in this approach; no additional functionalization process and reducing agents are required. We investigated in detail the influence of Au salt concentrations, annealing temperatures, and annealing times on the formation of NPs. Moreover, to better understand the characteristics of Au NPs inside a polymer medium, a simulation model was also proposed, allowing to investigate their optical properties and to compare with experimental results.

## 2. Materials and Methods

Gold(III) chloride trihydrate (≥99.9% trace metals basis)/Au salt with the chemical formula HAuCl${}_{4}$.3H${}_{2}$O was purchased from Sigma-Aldrich. SU-8 2000.5 (epoxy-based negative photoresist) was purchased from MicroChem Corporation.

Au NPs were fabricated inside a polymer film deposited on a glass substrate following the process shown in Figure 1. The general procedure follows three steps, all of them in air environment conditions: (i) Au salt was mixed with SU-8 2000.5 at different weight ratios (wt.%) by stirring for 20 minutes for complete dissolution. (ii) The nanocomposite metal/resist solution was then deposited on a glass substrate by spin-coating at 500 rpm for 5 s and then at 2000 rpm for 30 s. (iii) After that, thermal annealing treatment was carried out at different temperatures, between room temperature and 240 ${}^{{}^{\circ}}$C (accuracy ±1 ${}^{{}^{\circ}}$C), using a standard hot plate.

The nanocomposite film thickness was measured using a profilometer in a clean room, and ranged from 500 to 1000 nm. The plasmonic color can be observed by eye and by using a standard camera combined with an optical microscope. The plasmonic properties of the Au NPs in SU-8 resist were characterized by an ultraviolet–visible (UV–Vis) spectrometer. To evaluate Au NPs sizes, shapes, and distributions, the nanocomposite was dropped on a carbon-coated Cu grid and examined by a transmission electron microscope (TEM).

## 3. Experimental Results and Discussions

### 3.1. Mechanism

Generally, the process to form Au NPs needs to be implemented through complete reduction of Au precursor to induce nucleation and then particles. For example, Au NPs synthesis in SU-8 was previously carried out under light illumination, where SU-8 acts as a photoreduction agent and its polymerization effect induces the formation of nucleations and NPs [12,33]. In the case described here, there is no obvious reducing agent present (SU-8 exists but without light). We observed that the solvent of SU-8 resist plays an important role in the formation of Au NPs. Indeed, we tested different kinds of SU-8 resists, such as SU-8 2000.5 (more solvent) and SU-8 2002 and 2005 (less solvent), but only SU-8 2000.5 allows us to obtain Au NPs. The reason is that the solvent concentration of SU-8 2000.5 is higher than that of SU-8 2002 and 2005, and enough to assist the movement of the Au salt. Thus, the mechanism of the formation of Au NPs in SU-8 can be explained as follows. First, it was suggested that the thermal effect decomposes the Au salt to form Au${}^{{}^{\circ}}$ [34]. During the annealing process, the solvent evaporates, pushing Au${}^{{}^{\circ}}$ to form crystalline seeds and, consequently, Au NPs. We therefore refer to this fabrication method as a solvent-evaporation-assisted thermal annealing technique.

### 3.2. Dependence on Concentrations

The influence of the different concentrations of Au salt on the formation of Au NPs is shown in Figure 2. Five nanocomposite samples of SU-8 2000.5 mixed with various concentrations of the Au salt, from 1 wt.% to 10 wt.%, were annealed at 95 ${}^{{}^{\circ}}$C for 5 min in the dark. The formation of Au NPs is confirmed by different measurements: by observing the color change of the sample (Figure 2a), by recording absorption spectra (Figure 2b), and by TEM images (Figure 3). The color becomes darker for higher concentrations because more NPs are formed. The higher concentration also leads to an increase in the absorbance, which is illustrated in Figure 2b. Furthermore, the absorption peaks displayed both a redshift, from around 550 nm for 1 wt.% to about 570 nm for 10 wt.%, and a broadening of the peak width. This physical phenomenon is due to the contribution of Au NPs with bigger sizes, which will be explained more clearly in the simulation part. In addition, the peak width broadening with higher concentration was also suggested by the previous works [16,35] due to more polydispersed Au NPs.

Figure 3 shows TEM images and the size distribution of generated Au NPs after thermal treatments. It is found that the increase in the concentration of Au salt leads to the generation of more Au NPs (Figure 3a–c). Indeed, for low concentrations of Au salts (1 wt.% and 3 wt.%), the NPs are well separated and easily distinguished. However, for high Au salt concentrations, such as 7 wt.%, Au NPs become too dense. We also found that different shapes of Au NPs are observed, such as nanospheres, nanoprisms, nanorods, and nanotrapezoids (Figure 3d–g), as compared to the dominant spherical shape on a large scale. The domination of spherical or near-spherical particles could be explained by the lowest energy needed for the nanosphere formation, which is thermodynamically preferred [36,37]. For nonspherical shape or anisotropic growth, the formation probability is much lower because it requires more energy to fight against thermodynamics [37]. To more easily control the particular shape of Au NPs, surfactants and shape-directing reagents could be added to selectively choose the specific facets to grow, resulting in desired particle shapes [36,38]. In actuality, special shapes such as Au nanorods exhibit two absorption peaks: one is related to electron oscillation along the transverse direction, and the other is related to electron oscillation along the longitudinal direction [39]. However, in Figure 2, only one resonant peak is observed because the contribution of these special shape NPs is very small compared to the contribution of spherical-like shape NPs. In addition, we can see in Figure 3h that most Au NPs are quite small, with diameters around 10 nm, indicating that in the actual conditions, this method creates mainly small NPs. This can bring some advantages because the smaller the size of Au NPs, the higher the ratio between surface area and volume, resulting in more catalytic activity, which can be used for biosensor technology [40,41,42]. However, it is important to note that for all samples obtained with different Au salt concentrations, we found that some bigger Au NPs are also formed even in very small quantities, and are surrounded by mostly small NPs (see the center of Figure 3b).

### 3.3. Dependence on Annealing Temperatures

The effect of annealing temperature on the formation of Au NPs was also investigated. Several nanocomposite samples of SU-8 2000.5 mixed with 1 wt.% Au salt were prepared and annealed on a hot plate at different temperatures for the same annealing duration of 5 min. As can be seen from Figure 4a, the increase in annealing temperatures leads to the modification of color shades of the metallic/polymeric samples: changing from colorless to purple after heating, becoming darker when increasing the temperature until 150 ${}^{{}^{\circ}}$C, and then becoming faded. UV-Vis absorption spectra were recorded and are shown in Figure 4b. Again, for the sample without thermal treatment, no absorption peak is observed. When the annealing temperature is increased, Au NPs are formed, as indicated by the resonant plasmonic peaks at around 550 nm. When the annealing temperature is varied, the SPR amplitudes and positions vary accordingly. This can be explained by the fact that the increase of the annealing temperature leads to an increase of the SU-8 solvent evaporation rate, which affects the formation of Au NPs. In general, SPR peaks have a tendency to shift toward longer wavelengths, and their widths broaden when the annealing temperature increases. In fact, the acceleration of the solvent evaporation process helps the formation of Au NPs with bigger sizes, thus explaining the redshift and broadening peak width effects, respectively. However, when heating the nanocomposite samples at higher temperatures, from 180 ${}^{{}^{\circ}}$C to 240 ${}^{{}^{\circ}}$C, as seen in Figure 4a, Au NPs cannot be formed and the plasmonic effect is not clearly visible, because at this high temperature, the SU-8 solvent evaporates too fast and Au salt/nucleations do not have enough time to form Au NPs. Furthermore, at high temperatures, the polymerization of SU-8 resist can occur via an effect called thermal polymerization, resulting in a more rigid polymer matrix, which also prevents the formation of Au NPs. It should be noted that the thermal annealing process is also applied to samples prepared several days earlier, but neither the color of the sample nor the absorption peaks can be observed. This is because after waiting for a few days, the SU-8 solvent has already evaporated from the spin-coated samples. This again confirms our proposed mechanism that the solvent of the SU-8 resist plays an important role in the generation of Au NPs. In addition, heat treatment is important, because without it, no color is observed, meaning that no Au NPs were formed for the samples stored in a dark box and at room temperature for a few days. For these samples, the solvent was slowly and totally evaporated.

### 3.4. Dependence on Annealing Duration

Another factor that contributes to the change of Au NPs optical properties is annealing duration. Several samples of SU-8 2000.5 mixed with 1 wt.% Au salt were prepared and then placed on a hot plate preliminary heated to a stable temperature (95 ${}^{{}^{\circ}}$C). Individual samples were taken out after a chosen annealing duration. Figure 5a shows sample colors after experiencing different annealing times. The purple color illustrates the formation of Au NPs, which was also proved by SPR peaks, as shown in Figure 5b. When the annealing time increases, the peaks have a tendency to move to longer wavelengths. This is because a longer annealing time leads to a higher probability of forming bigger Au NPs. The result does not change much when the annealing duration is longer than 25 min, since the formation of Au NPs is achieved. At longer annealing times, a possible effect is thermal polymerization of SU-8 resist, as mentioned above, which may induce a solid polymer matrix and a slight change of its refractive index, resulting in a small change of Au NPs SPR peak.

## 4. Numerical Investigation of Au NPs inside a Polymer Medium

Thus far, we experimentally characterized the formation and optical properties of Au NPs within SU-8 resist. Since Au NPs show highly dispersed size distributions, to better understand the contribution of size effect to the resonance peaks of the nanocomposite samples, theoretical or numerical studies are needed. Indeed, there are many studies on the influence of a single metal NP size on the plasmonic effect based on Mie theory [43,44,45,46]. However, when considering the case of multiple Au NPs with different sizes being placed together inside a polymeric medium, there is no work that comprehensively explains the role of the NPs size in the characteristics of the resonance peaks. Here, we performed numerical simulations using a commercial three-dimensional finite-difference time-domain (FDTD) solver (Ansys Lumerical software) to study the optical properties of Au NPs in a polymeric medium. The FDTD method is recognized as a powerful numerical method to simulate the optical effect of metal NPs or nanostructures [47,48,49,50,51].

### 4.1. Simulation Model

First, TEM images were processed by a MATLAB image processing algorithm to extract the size distribution of Au NPs. Then, spherical Au particles with the same number and size distribution were generated inside the FDTD simulation region using a random number seeding function with a normal distribution. As mentioned, the dominant shape of NPs is sphere-like; therefore, we chose the shape of NPs as a sphere for simplicity without affecting the physical meaning. We note that these spherical Au NPs were randomly distributed on a 500 nm thick layer of SU-8 film on top of a glass substrate to be consistent with the experiment. We assume that the refractive indices of SU-8 and glass are 1.58 and 1.48, respectively, in visible range. The dielectric constant for Au was taken from the experimental data of [52]. The exciting light source is a total-field scattered-field (TFSF) source with a wavelength range between 400 nm and 800 nm to match the operating bandwidth of the UV-Vis spectrometer. An override mesh region was added to the whole TFSF source region with a mesh size of 0.5 nm to ensure the convergence of the simulation results. The boundary conditions are perfectly matched layers, which allows outgoing waves from the inside of a computational region to be strongly absorbed without being reflected back. The absorbance was calculated from an analysis group, which is located inside the TFSF source but outside the Au NPs.

### 4.2. Simulation Results

#### 4.2.1. Effect of Particle Size Distribution

The experimental results show that Au NPs have polydisperse size distribution, with most of the small NPs under 20 nm in diameter and a small fraction of bigger NPs with sizes ranging from 20 to 100 nm, depending on the sample and the TEM image acquisition location. Since the size distribution of Au NPs is clearly divided into two regions, to evaluate the size effect, we simulate three separate cases, as follows. In the first case, only Au NPs with sizes smaller than 20 nm are considered. In the second case, we consider only three Au NPs with a size of 50 nm, representing big particles. In the third case, a combination of the two previous cases is investigated. Results are shown in Figure 6. As can be clearly seen, for the first case, when the simulation region only contains small Au NPs, the absorbance value is very low compared to the absorbance value of the second case, where three big Au NPs are simulated. As a result, for the third case, the absorption spectrum is almost identical to the second case and is mainly made of three big Au NPs contributions. It is also worth noting that the peak experiences a redshift when adding three big Au NPs into the first case, which shows that both the amplitude and the position of the resonance peaks are determined mainly by big Au NPs. This is understandable since a particle with a diameter of 50 nm has a scattering cross-section as well as a volume many times larger than for a particle with a size of 10 nm. This is also indicated in Mie theory when the absorption coefficient and peak position depend strongly on the radius of spherical Au NPs [53].

#### 4.2.2. Effect of Particle Size and Comparison with Experimental Results

In the previous section, the big Au NPs are demonstrated as the main contributors to the amplitude and position of the absorption peaks. We also conducted simulations for cases with varying sizes of big NPs and compared them with the experiment to try to explain the primary mechanism that affects the absorption peaks. Figure 7a shows the simulated absorption spectra of five simulation regions, which contain both small (10 nm) Au NPs and three big Au NPs having the size from 20 nm to 60 nm. As can be clearly seen, when the size of big Au NPs increases, the resonance peak experiences redshift, increased amplitude value, and widening. When comparing them to the experiment results in Figure 2b, the absorption spectra follow the same trend when increasing Au salt concentration. This trend is also demonstrated by comparing the position of the resonance peaks between simulation and experiment, as shown in Figure 7b. This implies that, in practice, when increasing the concentration of Au salt, Au NPs are generated with a bigger size even though the number of particles smaller than 20 nm still dominates. In actuality, due to technical limitations, we can only observe Au NPs in some regions on a carbon-coated Cu grid by using TEM where the thickness of the film is very thin (≈500 nm). Therefore, simulation results are needed to gain a better understanding of the nanocomposite samples. We note that the experimental spectral peaks are always wider than the simulated ones. This is because when using the UV-Vis spectrometer, the illuminated region on the sample is very large (≈mm). We therefore measured signals from several big particles (main contributors) of different sizes, leading to a widening of the absorption spectrum peak, while for simulations, only three big Au NPs were considered.

## 5. Conclusions

In this study, a simple, low-cost, and fast method of thermal annealing for directly fabricating Au NPs inside a polymer matrix was demonstrated. Firstly, we heated the nanocomposite sample of an SU-8 resist mixed with Au salt in suitable conditions to grow Au NPs. The formation of Au NPs was explained by the thermal decomposition and solvent evaporation at suitable annealing temperatures and duration. Excellent Au NPs in SU-8 resist were obtained by optimum conditions: (i) concentrations of Au salt of 1–3 wt.%; (ii) annealing temperatures of 90–100 ${}^{{}^{\circ}}$C; and (iii) annealing duration of 1–5 min. The optical properties of Au NPs were characterized by different methods, such as optical and electronic microscopes and UV-Vis spectrometers. Secondly, the FDTD simulation model was first developed to investigate the optical properties of Au NPs inside polymer medium. Thanks to the simulation results, the size of Au NPs was recognized as an important contributor to the resonance peaks when considering many Au NPs with different sizes distributed in the space. Different applications of this Au NPs/SU-8 nanocomposite could be exploited, such as for data storage, plasmonic/photonic devices, and surface-enhanced Raman scattering (SERS) spectroscopy. This approach suggests that this thermal effect could be used by other setups, such as laser systems to directly fabricate metallic NPs within the polymer with potential applications to plasmonic and metamaterial structures.

## Figures and Tables

**Figure 1 polymers-15-00016-f001:**
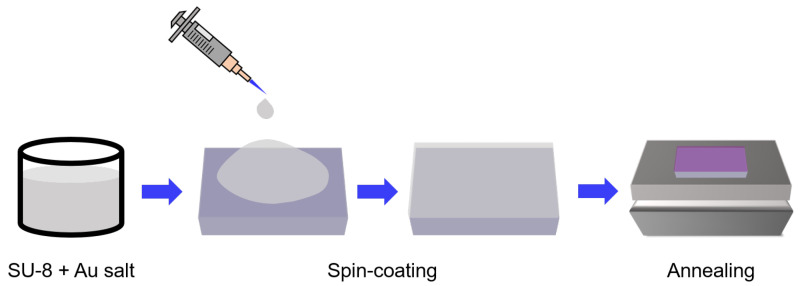
Fabrication procedure of Au NPs in polymer film by a solvent-evaporation-assisted thermal annealing technique.

**Figure 2 polymers-15-00016-f002:**
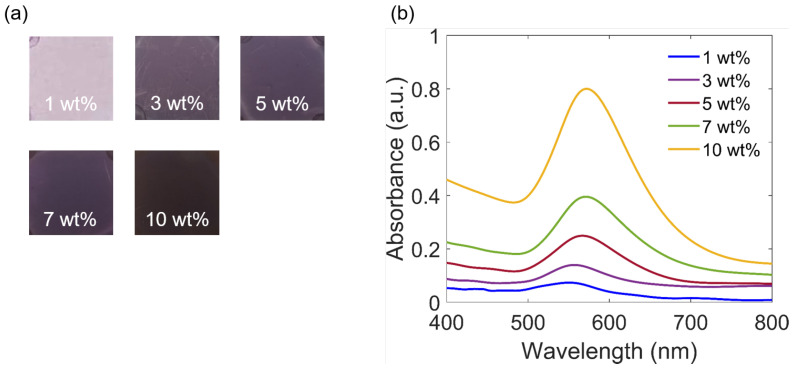
The formation of Au NPs depends on the concentrations of Au salt: (**a**) Direct smartphone pictures of the nanocomposite samples, obtained by SU-8 mixed with different concentrations of Au salt after annealing at 95 ${}^{{}^{\circ}}$C in 5 min. (**b**) The corresponding absorption spectra indicating the plasmonic effects of Au NPs inside SU-8 thin film.

**Figure 3 polymers-15-00016-f003:**
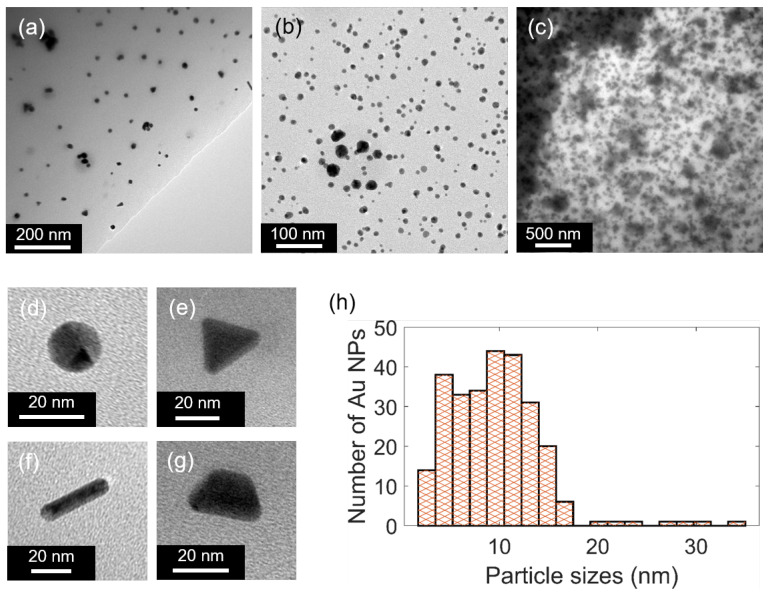
TEM images of the samples with different concentrations of Au salt: (**a**) 1 wt.%, (**b**) 3 wt.%, and (**c**) 7 wt.% after annealing at 95 ${}^{{}^{\circ}}$C in 5 min. (**d**–**g**) The special shapes of generated Au NPs after thermal treatment. (**h**) The size distribution of Au NPs for 3 wt.% Au salt.

**Figure 4 polymers-15-00016-f004:**
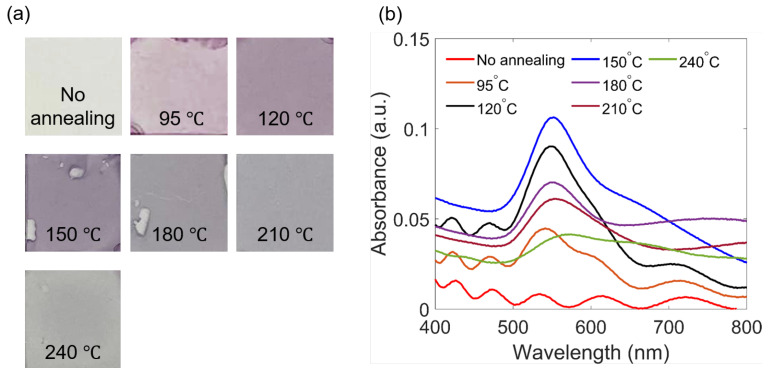
The temperature dependence on the formation of Au NPs: (**a**) Direct smartphone pictures of the nanocomposite samples of SU-8 mixed with 1 wt.% Au salt, obtained by thermal treatment at different temperatures. (**b**) The corresponding SPR spectra.

**Figure 5 polymers-15-00016-f005:**
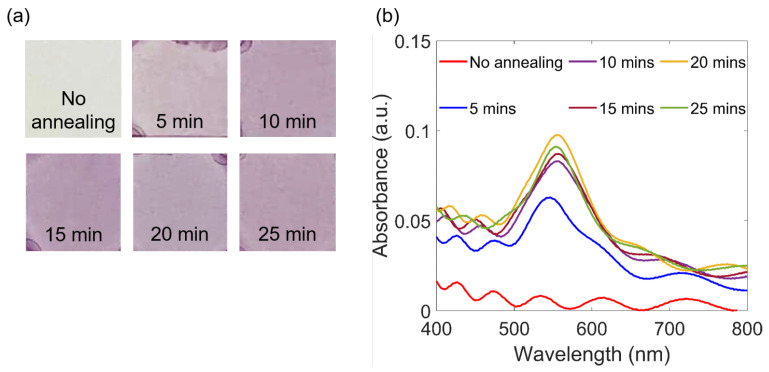
Formation of Au NPs depends on different annealing times: (**a**) Direct smartphone pictures of the nanocomposite samples of SU-8 mixed with 1 wt.% Au salt, obtained by annealing method at 95 ${}^{{}^{\circ}}$C for different durations. (**b**) The corresponding SPR spectra.

**Figure 6 polymers-15-00016-f006:**
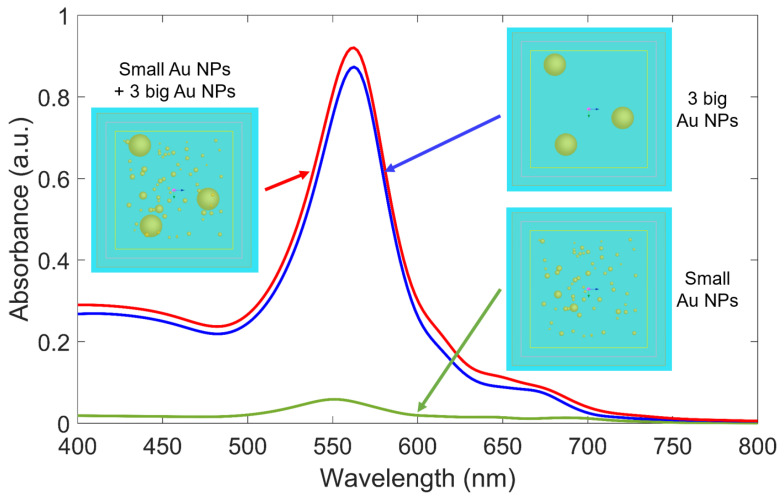
Absorption spectra of Au NPs in three cases: (i) only small Au NPs with the size of around 10 nm, (ii) only 3 big Au NPs with the size of 50 nm, and (iii) the combination between many numbers of small Au NPs and 3 big Au NPs. The three insets show the top view of the corresponding three simulation models using FDTD. The refractive indices of surrounding medium (SU-8) and glass substrate are 1.58 and 1.48, respectively.

**Figure 7 polymers-15-00016-f007:**
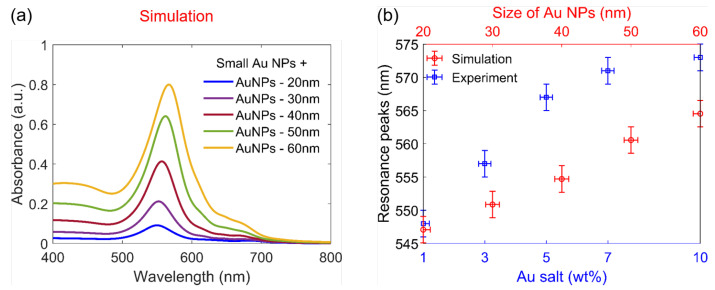
Absorption spectra of Au NPs: (**a**) Theoretical calculation of the regions containing Au NPs with different sizes from 20 nm to 60 nm surrounded by many small (10 nm) Au NPs. (**b**) Comparison of resonance peak positions between simulation and experiment. The error bars indicated for each point are as follows: ±1 nm for Au NPs size; ±0.2 wt.% for Au salt concentration; and ±2 nm for SPR peak position.

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
