# Peer review of "Direct Synthesis of Gold Nanoparticles in Polymer Matrix"

_polymers, 2022, doi:10.3390/polym15010016_

Round 1

Reviewer 1 Report

In this work, Au NPs were synthesized in the SU-8 photoresist by thermal reduction of gold. A detailed characterization of the obtained NPs in terms of morphology and optical effects was realized.

The manuscript is well written, and data is presented properly. A demonstration of the practical application of the obtained polymer dopped with AuNPs should have been presented in order to increase the interest to the readers e.g.: SERS analysis.

·         The solvent’s influence on the NP formation is based on its concentration in the SU-8 photoresist? All the tested photoresists were based on the same solvent?

·         Was there any correlation between the used concentration of the Au salt and the differently shaped Au-nanostructures obtained (nanoprism, nanorod, and nanotrapezoid), or they were present in all cases?

·         Did the authors use a controlled environment during the NP synthesis e.g.: nitrogen instead of air?

Hence, paper may be accepted for the publication.

Author Response

Question/comment 1:

The manuscript is well written, and data is presented properly. A demonstration of the practical application of the obtained polymer doped with AuNPs should have been presented in order to increase the interest to the readers e.g.: SERS analysis.

• Answer 1:
Thank you for this comment. Effectively, we would love to demonstrate some applications of these nanocomposite samples. We have mentioned several potential applications in the conclusion of the manuscript. The applications demonstration is however out of scope of this manuscript, which is already quite long and complete. We believe that different applications will be demonstrated and published in near future.

Question/comment 2:

The solvent’s influence on the NP formation is based on its concentration in the SU-8 photoresist? All the tested photoresists were based on the same solvent?

• Answer 2:
Yes. The SU-8 photoresist solvent concentrations definitely affect the plasmonic NPs formation. All the tested photoresists were based on the same solvent, which is cyclopentanone.

Question/comment 3:

Was there any correlation between the used concentration of the Au salt and the differently shaped Au-nanostructures obtained (nanoprism, nanorod, and nanotrapezoid), or they were present in all cases?

• Answer 3:
We have observed various shapes of Au NPs when using different concentrations of Au salt. However, we didn’t notice any correlation between the used concentration of the Au salt and the differently shaped Au NPs. The main formed shape of NPs is spherical-like.

Question/comment 4:

Did the authors use a controlled environment during the NP synthesis e.g.: nitrogen instead of air?

• Answer 4:
Thank you for this question. In our experiment, we have proposed a simplest experimental condition to synthesis Au NPs in an air environment. The effect of environmental atmosphere may affect the formation of An NPs, and it will be studied in future work.

Reviewer 2 Report

REVIEW

to Quang Truong Pham, Gia Long Ngo, Xuan An Nguyen, Chi Thanh Nguyen, Isabelle Ledoux-Rak, and Ngoc Diep Lai “Direct synthesis of gold nanoparticles in polymer matrix”

The purpose of the scientific study presented by the authors is to improve the method of forming the SU-8/AuNPs composite with plasmonic properties. A mechanism for the direct formation of AuNPs directly in the SU-8 matrix has been proposed, and their sizes and shape of nanoparticles have been determined. The dependence of these parameters, as well as the optical properties, on the concentration of the HAuCl4 salt, temperature, and annealing time has been established. To better understand the characteristics of AuNPs inside a polymer medium, a simulation model was also proposed, allowing to investigate their optical properties and to compare with experimental results. The presented article will be of interest to researchers working in this field.

However, the article needs minor editing.

1. The authors tested various types of SU-8 resists, such as SU-8 2000.5 (more solvent), and SU-8 2002 and 2005 (less solvent). They found that only SU-8 2000.5 produced AuNPs. However, there is no specific explanation for this fact in the manuscript. It is necessary to explain what exactly this selective behavior of SU-8 is connected with. What is the reason for this effect?

2. Since the reduction of HAuCl4 3 H2O to AuNPs occurs when heated in the range from room temperature to 240 °C, it is necessary first of all to indicate the temperature stability of SU-8 (glass transition, melting, decomposition temperature) in order to justify the possibility of using SU-8 in the selected range temperature annealing during the formation of AuNPs..

3. In the Conclusion (lines 257-259), the authors state that the optimum conditions for obtaining SU-8/AuNPs composites with the best optical properties were found. However, this statement is not confirmed by anything. It is necessary to enter specific data of these optimum conditions.

After the response to these comments, the article can be accepted for publication.

Author Response

Question/comment 1:

The authors tested various types of SU-8 resists, such as SU-8 2000.5 (more solvent), and SU-8 2002 and 2005 (less solvent). They found that only SU-8 2000.5 produced AuNPs. However, there is no specific explanation for this fact in the manuscript. It is necessary to explain what exactly this selective behavior of SU-8 is connected with. What is the reason for this effect?

• Answer 1:
Thank you for this question. As mentioned in our manuscript, we have already tested with different kinds of SU8 resist, such as SU8-2002, SU8-2005, and SU8-2000.5. However, only SU8-2000.5 resist allowed us to produce Au NPs with good quality. The mechanism of the formation was explained by the solvent evaporation-assisted thermal effect in section 3.1. To be more clear, in revised manuscript, we add one more sentence in this section: The reason is that the solvent concentration of SU-8 2000.5 is higher than that of SU-8 2002 and 2005, and enough to assist the movement of the Au salt.

Question/comment 2:

Since the reduction of HAuCl4.3H2O to AuNPs occurs when heated in the range from room temperature to 240 °C, it is necessary first of all to indicate the temperature stability of SU-8 (glass transition, melting, decomposition temperature) in order to justify the possibility of using SU-8 in the selected range temperature annealing during the formation of AuNPs.

• Answer 2:
Thank you for this remark. SU-8 is a commercial product (MicroChem Corporation). The thermal stability and glass transition temperature of SU-8 are 315°C and 210°C, respectively. Therefore, all tested temperatures are below the thermal stability temperature, which means that there is no degradation of SU-8 during the formation of Au NPs. Some discussions of the temperature effect on the polymerization of SU-8 resist are shown in section 3.3 (Dependence on annealing temperatures).

Question/comment 3:

In the Conclusion (lines 257-259), the authors state that the optimum conditions for obtaining SU- 8/AuNPs composites with the best optical properties were found. However, this statement is not confirmed by anything. It is necessary to enter specific data of these optimum conditions.

• Answer 3:
Thank you for this comment. In revised manuscript, we have optimized the text and changed this sentence: Excellent Au NPs in 257 SU-8 resist were obtained by optimum conditions: i) concentrations of Au salt of 1–3 wt.%; ii) annealing temperatures of 90- 100°C; and iii) annealing duration of 1–5 minutes. The optical properties of Au NPs were characterized by different methods, such as optical and electronic microscopes, UV-VIS spectrometers.

Reviewer 3 Report

NICE WORK. BUT, CONCLUSION NEEDS TO BE IMPROVED

Author Response

Nice work. But, conclusions need to be improved.

• Answer:

Thank you for this suggestion. In revised manuscript, we have optimized the conclusion by modifying and adding new sentences: Excellent Au NPs in 257 SU-8 resist were obtained by optimum conditions: i) concentrations of Au salt of 1–3 wt.%; ii) annealing temperatures of 90-100°C; and iii) annealing duration of 1–5 minutes. The optical properties of Au NPs were characterized by different methods, such as optical and electronic microscopes, UV- VIS spectrometers......Different applications of this Au NPs/SU-8 nanocomposite could be exploited, such as data storage, plasmonic/photonic devices, as well as surface-enhanced Raman scattering (SERS) spectroscopy.